# NMNAT1 inhibits axon degeneration via blockade of SARM1-mediated NAD+ depletion

Yo Sasaki[1]*, Takashi Nakagawa[2], Xianrong Mao[1], Aaron DiAntonio[3]*, Jeffrey Milbrandt[1]*

[1]Department of Genetics, Washington University School of Medicine, Saint Louis, United States; [2]Frontier Research Core for Life Sciences, University of Toyama, Toyama, Japan; [3]Department of Developmental Biology, Washington University School of Medicine, Saint Louis, United States

**Abstract** Overexpression of the NAD+ biosynthetic enzyme NMNAT1 leads to preservation of injured axons. While increased NAD+ or decreased NMN levels are thought to be critical to this process, the mechanism(s) of this axon protection remain obscure. Using steady-state and flux analysis of NAD+ metabolites in healthy and injured mouse dorsal root ganglion axons, we find that rather than altering NAD+ synthesis, NMNAT1 instead blocks the injury-induced, SARM1-dependent NAD+ consumption that is central to axon degeneration.

*For correspondence: ysasaki@ genetics.wustl.edu (YS); diantonio@wustl.edu (AD); jmilbrandt@wustl.edu (JM)

## Introduction

Axon loss following injury and disease (Wallerian degeneration) is a regulated process of axon self-destruction (*Coleman, 2005*; *Gerdts et al., 2016*; *Wang et al., 2012*). The discovery of the Wallerian degeneration slow (Wlds) protein, which dramatically delays axon degeneration after injury, raised hopes that blocking this process would be useful in the treatment of neurological disorders (*Conforti et al., 2014*; *Mack et al., 2001*). The Wlds protein blocks axon degeneration by mislocalizing the nuclear nicotinamide adenine dinucleotide (NAD+) biosynthetic enzyme NMNAT1 into axons, thereby substituting for the loss of the labile axon maintenance factor NMNAT2 (*Araki et al., 2004*; *Babetto et al., 2010*; *Gilley et al., 2010*; *Sasaki et al., 2010*). The mechanism by which NMNAT enzymes block axon degeneration is unknown. These enzymes synthesize NAD+ from nicotinamide mononucleotide (NMN) and ATP, so two candidate mechanisms for how loss of NMNAT2 triggers axon degeneration have emerged: (1) the loss of NAD+ or (2) the accumulation of NMN. The striking axonal protection provided by neuronal expression of *Escherichia coli* NMN deamidase, which reduces NMN levels by converting it to nicotinic acid mononucleotide (NaMN), supports the model that NMN accumulation triggers axon degeneration (*Di Stefano et al., 2015*). Here we investigate NAD+ metabolism in healthy and injured axons through the measurement of steady-state metabolite levels and via the analysis of NAD+ metabolite synthesis and consumption (i.e. flux analysis). Surprisingly, we find that neither NAD+ loss nor NMN accumulation trigger axon degeneration. Instead, both NMNAT1 and NMN deamidase prevent axon degeneration via blocking the injury-induced NAD+ consumption that occurs following activation of the axodestructive molecule SARM1 (*Gerdts et al., 2013*, *2015*; *Osterloh et al., 2012*).

To investigate whether NAD+ loss or NMN accumulation triggers axon degeneration, we assessed injury-induced axon degeneration and NAD+ metabolite levels in cultured DRG neurons in which the NAD+ biosynthetic pathway was perturbed at various steps (*Figure 1a*). To raise intracellular NMN levels we expressed the NMN biosynthetic enzyme NAMPT or applied nicotinamide

**Figure 1.** Various modes of axonal protection mediated by the manipulation of NAD$^+$ synthesis pathways. (a) Diagram of mammalian nicotinamide adenine dinucleotide (NAD$^+$) biosynthesis pathways. Nicotinamide (Nam) is the major NAD$^+$ precursor in mammals. It is converted to nicotinamide mononucleotide (NMN) by the NAMPT (nicotinamide phosphoribosyltransferase) enzyme, and then to NAD$^+$ by the NMNAT (nicotinamide mononucleotide adenylytransferase) enzymes. There are three isoforms of NMNAT in mammalian cells; NMNAT1, 2, and 3. NMN deamidase is a bacterial enzyme that converts NMN to nicotinic acid mononucleotide (NaMN). NMN synthetase is also a bacterial enzyme that performs the reverse reaction and converts NaMN to NMN. NaMN is converted to nicotinic acid adenine dinucleotide (NaAD) by NMNAT1, 2, or 3. NaAD Is converted to NAD$^+$ by NAD synthetase. Nicotinamide riboside (NR) is an alternative NADprecursor that is converted to NMN by NRK1 or NRK2 (nicotinamide riboside kinase enzymes). FK866 is a chemical inhibitor of NAMPT enzymatic activity. (b) Representative images of DRG axons at 24 hr after axotomy. DRG neurons were infected with lentivirus expressing mCherry protein (control), cytNMNAT1, NMN deamidase (NMN DD), NAMPT, or NRK1 at four days prior to axotomy. Nicotinamide riboside (NR, final 100 μM in the culture medium) was added to control or NRK1 expressing neurons at 24 hr prior to axotomy and FK866 (final 100 nM in the culture medium) was added at the time of axotomy. (c) Axon degeneration under various manipulations of NAD$^+$ biosynthesis pathways was quantified using an axon degeneration index. Data show mean ± s.d., one-way ANOVA F(31,256) = 149.5, p<$2 \times 10^{-16}$. *p<0.005 and **p<$1 \times 10^{-7}$ denote a significant difference from control DI before axotomy with Holm-Bonferroni multiple comparison (*Figure 1—source data 1*, n = 9 for each time point).

The following source data and figure supplement are available for figure 1:

**Source data 1.** Axonal degeneration index at 0, 9, 24, 48, and 72 hr post axotomy in the presence of various NAD$^+$ biosynthesis manipulations (*Figure 1c*).

**Figure supplement 1.** Lentivirus mediated expression of NAD$^+$ biosynthesis enzymes in axons.

riboside (NR) to neurons expressing NRK1, a nicotinamide riboside kinase that converts NR to NMN. To reduce NMN levels we treated neurons with the NAMPT inhibitor FK866 or expressed *E. coli* NMN deamidase (*Di Stefano et al., 2015*). We also tested the potent axoprotective cytoplasmic version of NMNAT1 (cytNMNAT1), which appears to substitute in axons for the short-lived NMNAT2 (*Gilley et al., 2010*, *Sasaki et al., 2009a*). The expression of these enzymes in axons was confirmed by Western blotting (*Figure 1—figure supplement 1*). Using our automated image analysis axon degeneration assay, and consistent with previous reports, all of these manipulations delayed axon

degeneration although to dramatically different extents (*Figure 1b,c* and *Figure 1—source data 1*; *Di Stefano et al., 2015*; *Sasaki et al., 2006*, *2009b*). FK866 and NR treatment provided only modest protection, delaying axon degeneration by ~6 to 9 hr. Neurons expressing NRK1 and treated with NR or expressing NAMPT showed strong axon protection for 24 to 48 hr. The strongest axonal protection was found in neurons expressing cytNMNAT1 or NMN deamidase, which both block axon degeneration for at least 3 days after axotomy.

To correlate axon degeneration with levels of NAD$^+$ metabolites, we measured baseline NMN, NaMN, NAD$^+$, and nicotinic acid adenine dinucleotide (NaAD) from DRG neurons using LC-MS/MS before axonal transection. As previously described, FK866 led to a slow decline in both cellular NAD$^+$ and NMN before the axons begin to fragment (*Figure 2a* and *Figure 2—source data 1*; *Di Stefano et al., 2015*). In contrast, NAMPT expression or pre-incubation with NR in the presence of NRK1 significantly increased both NMN and NAD$^+$ levels. NMN deamidase significantly increased the levels of NaMN and NaAD while dramatically reducing NMN and NAD$^+$ levels in DRG neurons (*Figure 2b,c* and *Figure 2—source data 1*). Despite reducing NAD$^+$ levels to 11 ± 7% of control, NMN deamidase-expressing neurons showed no signs of cell death or axon degeneration, but instead displayed potent axonal protection (*Figure 3c* and *Figure 3—source data 1*) as previously reported (*Di Stefano et al., 2015*). The protection afforded by NMN deamidase was equivalent to that observed in neurons expressing cytNMNAT1 (*Figure 3c* and *Figure 3—source data 1*), however baseline levels of NMN and NAD$^+$ were normal in neurons expressing cytNMNAT1 (*Figure 2b,c* and *Figure 2—source data 1*). Similarly, SARM1-deficient neurons had baseline levels of NMN, NAD$^+$, NaMN, and NaAD that were equivalent to those of wildtype neurons (*Figure 2—figure supplement 1* and *Figure 2—source data 2*, *Gerdts et al., 2015*). From these studies, it is clear that robust axonal protection can be observed in neurons that maintain low levels of NMN and NAD$^+$ (NMN deamidase), normal levels of NMN and NAD$^+$ (cytNMNAT1 or SARM1 knockout), or high levels of NMN and NAD$^+$ (NAMPT or NRK1 + NR).

The above results demonstrate that changes in baseline levels of NMN and NAD$^+$ are insufficient to trigger axon degeneration, so we next assessed changes in axonal NAD$^+$ metabolites after axotomy (*Figure 2d,e* and *Figure 2—source data 1*). We plated neurons in such a manner that we could harvest axons separate from cell bodies and thereafter use LC-MS/MS to measure axon-specific metabolite levels. These studies showed a dramatic decline in NAD$^+$ levels after injury in control axons, consistent with previous reports (*Di Stefano et al., 2015*; *Gerdts et al., 2015*; *Wang et al., 2005*). We also found a ~2-fold increase in the levels of NMN within the first two hours after injury. NMN levels declined back to those observed in uninjured axons by six hours post-axotomy. A previous report showed that sciatic nerve transection causes a continuous increase in nerve NMN concentration when it is normalized to the nerve adenylate pool (*Di Stefano et al., 2015*). We confirm this finding using purified axon preparations, however, we demonstrate that this is due to a rapid decline in the adenylate pool rather than an increase in NMN levels (*Figure 2—figure supplement 2* and *Figure 2—source data 3*). We next assessed NAD$^+$ metabolites in axons that are robustly protected from axotomy-induced degeneration to determine whether these treatments are correlated with a specific metabolite profile. In cells expressing the strongly axoprotective cytNMNAT1, axonal NAD$^+$ and NMN were comparable to baseline levels in uninjured axons of control neurons and these metabolite levels did not change after axotomy. Expression of NMN deamidase reduced axonal NAD$^+$ and NMN compared with control axons, and these metabolite levels did not change after axotomy. In contrast, axons from neurons expressing NAMPT had high levels of both NAD$^+$ and NMN at baseline and at six hours after axotomy. In axons from neurons expressing NRK1 and treated with NR there was a dramatic increase of NAD$^+$ and NMN levels at baseline. After injury, these axons showed a robust decrease in NAD$^+$ but further increases in NMN. Indeed, NMN levels in axons from these neurons were 16-fold higher than in control axons (0.16 ± 0.04 μM vs. 0.01 ± 0.001 μM, *Figure 2e* and *Figure 2—source data 1*), yet these axons were strongly protected from degeneration (>48 hr). These results demonstrate that even within injured axons, elevated levels of NMN do not promote axon degeneration but rather can be compatible with robust axonal protection.

While these data are inconsistent with the hypothesis that elevated NMN induces axon degeneration, the finding that NMN deamidase both potently protects axons and leads to low levels of NMN does support this hypothesis (*Di Stefano et al., 2015*). Hence, we sought to determine whether maintaining low NMN levels is necessary for the axoprotective function of NMN deamidase. We

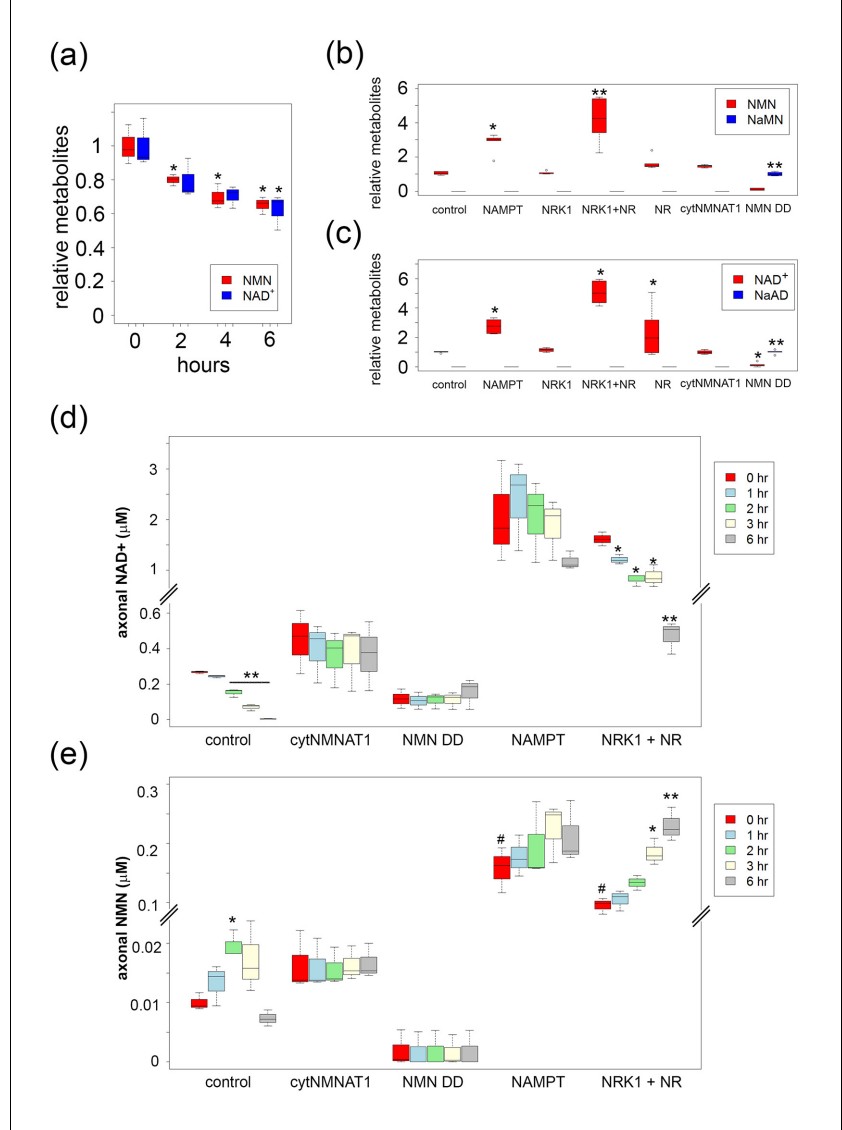

**Figure 2.** High levels of NMN are not sufficient to induce axon degeneration. (a) Intracellular NMN and NAD$^+$ levels relative to control (0 hr) were quantified at 2, 4, and 6 hr after 100 nM FK866 addition using LC-MS/MS. Data show the first and third quartile (box height) and median (line in the box) ± 1.5 time interquartile (details in the method), one-way ANOVA F(3,8) = 12.9, p=0.00197 for NMN; F(3,8) = 6.541, p=0.0152 for NAD$^+$. *p<0.05 denotes significant difference from metabolite levels at 0 hr with Holm-Bonferroni multiple comparison (*Figure 2—source data 1*, n=3). (b, c) Intracellular NMN and nicotinic acid mononucleotide (NaMN) levels (b), and NAD$^+$ and nicotinic acid adenine dinucleotide (NaAD) levels (c) relative to control (for NMN, NAD$^+$) or relative to NMN DD-expressing neurons (for NaMN, NaAD) were measured using LC-MS/MS. Data show the first and third quartile (box height) and median (line in the box) ± 1.5 time interquartile (details in the method), one-way ANOVA F(6, 35) =369.8, p<2× 10$^{-16}$ for NaMN; F(6, 35) = 29.92, p=2.14×10$^{-12}$ for NMN; F(6, 35) = 537, p<2×10$^{-16}$ for NaAD; F(6, 35) = 33.69, p=3.83× 10$^{-13}$ for NAD. *p<0.05, **p<1×10$^{-4}$ denote a significant difference from control metabolite levels with Holm-Bonferroni multiple comparison (*Figure 2—source data 1*, n = 6). (d, e) Axonal NAD$^+$ (d) and NMN (e) were quantified at 0, 1, 2, 3, and 6 hr post axotomy using LC-MS/MS. Axonal metabolites were collected from transected axons at the indicated time after axotomy. Data show the first and third quartile (box height) and median (line in the box) ± 1.5 time interquartile (details in the method), one-way ANOVA F(4,10) = 193.6, p=2×10$^{-9}$ for control NAD$^+$; F(4,10) = 6.682, p=6.694×10$^{-3}$ for control NMN; F(4,10) = 28.87, p=1.87×10$^{-5}$ for NRK1 NR, NAD$^+$; F(4,10) = 24.49, p=3.78×10$^{-5}$, for NRK1+ NR NMN. There are no significant metabolite changes after axotomy for NAMPT, cytNMNAT1, or NMN DD expressing cells (F(4, 10) = 1.066, p=0.422 for NAMPT NAD$^+$, F (4,10) = 0.893, p=0.503 for NAMPT NMN, F(4,10) = 0.127, p=0.969 for cytNMNAT1 NAD$^+$, F(4,10)=0.366, p=0.828 for cytNMNAT1 NMN, F(4,10)=0.343, p=0.843 for NMN DD NAD$^+$, F(4,10) = 0.004, p=1 for NMN DD NMN).
*Figure 2 continued on next page*

*Figure 2 continued*

*p<0.05, **p<1×10$^{-4}$ denote significant difference from axonal metabolite levels at 0 hr post axotomy with Holm-Bonferroni multiple comparison (*Figure 2—source data 1*, n = 3). # p<0.05 denotes significant difference of baseline NAD$^+$ or NMN before axotomy compared with control; One-way ANOVA with Holm-Bonferroni multiple comparison, $F_{(4, 10)}$ = 10.8, p=0.0012 for NAD$^+$ and $F_{(4,10)}$ = 41.44, p=3.1×10$^{-6}$ for NMN (*Figure 2—source data 1*, n = 3).

The following source data and figure supplements are available for figure 2:

**Source data 1.** Steady state cellular NMN, NaMN, NAD$^+$, and NaAD levels in the absence or presence of various NAD$^+$ biosynthesis manipulations (*Figure 2a* (control or FK866) and *Figure 2b,c* (control or various NAD$^+$ enzyme expressing neurons)).

**Source data 2.** Steady state NMN, NaMN, NAD$^+$, and NaAD levels in wild-type or SARM1 KO DRG neurons (*Figure 2—figure supplement 1*).

**Source data 3.** Steady state axonal total adenylate levels (ATP + ADP + AMP) and NMN normalized by total adenylate levels at 0, 1, 2, 3, 6 hr post axotomy (*Figure 2—figure supplement 2*).

**Figure supplement 1.** SARM1 KO neurons has normal NAD+ related metabolites.

**Figure supplement 2.** Axonal NMN and adenylate pool after axotomy.

---

approached this problem by attempting to maintain normal NMN levels in neurons expressing *E. coli* NMN deamidase and then assessing whether axons now degenerate after axotomy. To this end, we utilized an enzyme from *Francisella tularensis* called NMN synthetase that catalyzes the conversion of NaMN to NMN (i.e., the reverse of the reaction catalyzed by *E. coli* NMN deamidase) (*Figure 1a*, *Figure 1—figure supplement 1*). When we expressed NMN synthetase alone in neurons no changes in any of the measured metabolites were observed (*Figure 3a,b* and *Figure 3—source data 1*), and injured axons from these neurons degenerated normally (*Figure 3c* and *Figure 3—source data 1*). However, when NMN synthetase was co-expressed along with NMN deamidase, the normal NMN deamidase-catalyzed reductions in NMN and NAD$^+$ as well as the increases in NaMN and NaAD are largely abolished (*Figure 3a,b* and *Figure 3—source data 1*). Furthermore, the addition of the NMN precursor NR to neurons co-expressing *F. tularensis* NMN synthetase and *E. coli* NMN deamidase raised axonal NMN well above control levels (*Figure 3a* and *Figure 3—source data 1*). Despite this elevation in NMN levels under these conditions, NMN deamidase continued to provide robust axon protection (*Figure 3c* and *Figure 3—source data 1*). In addition, we also found that protection of injured axons did not correlate with levels of NAD$^+$, NaMN, or NaAD. Hence, neither a reduction in NMN nor the accumulation of NaMN/NaAD is necessary for NMN deamidase mediated axonal protection. These findings are inconsistent with the hypothesis that the axonal NMN level is a driver of axon degeneration.

Following axotomy, axonal NAD$^+$ levels decline. This decline could be due to loss of NAD$^+$ biosynthesis via degradation of the labile axonal NMNAT isoform NMNAT2 (*Gilley et al., 2010*), and/or to an increase in NAD$^+$ degradation due to activation of the pro-degenerative molecule SARM1 (*Gerdts et al., 2015*). To interrogate the processes of NAD$^+$ synthesis and degradation independently, we developed an axonal NAD$^+$ flux assay. The nicotinamide (Nam) in the DRG culture medium was largely replaced with deuterium-labeled nicotinamide (D4-Nam: 2,4,5,6-deutrium Nam, 300 μM) to achieve a 10-fold excess over the original Nam concentration in culture medium). The addition of up to 1 mM of this NAD$^+$ precursor did not alter the time course of axon degeneration (*Figure 4—figure supplement 1* and *Figure 4—source data 2*) and allowed us to identify its labeled-NAD$^+$ derivative using LC-MS/MS. We added labeled Nam at time zero, and followed its conversion by quantifying non-labeled (light) and labeled (heavy) NAD$^+$ in axons using LC-MS/MS at various times after addition. Conversion of the NAD$^+$ pool from light to heavy was observed without delay, demonstrating that D4-Nam replaces Nam for NAD$^+$ synthesis (*Figure 4—figure supplement 2*). The increase in heavy NAD$^+$ following D4-Nam addition gives a measure of the NAD$^+$ biosynthetic rate, while the loss of light NAD$^+$ reflects the NAD$^+$ consumption rate. In uninjured axons

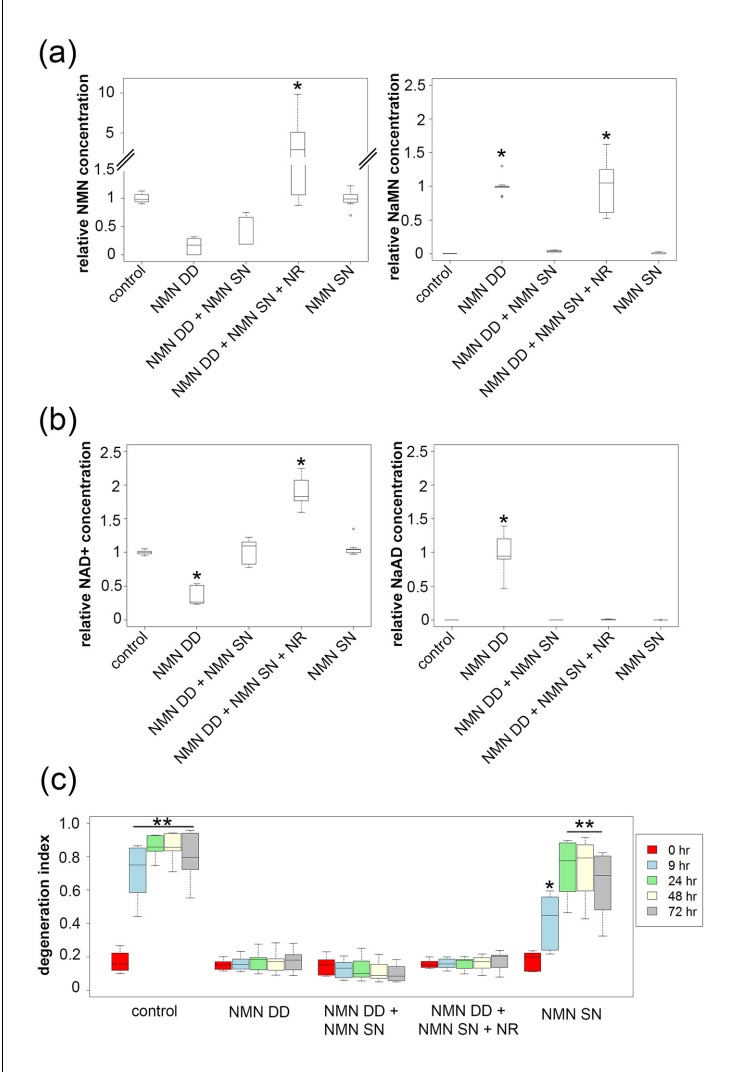

**Figure 3.** NMN deamidase does not protect axons by reducing NMN levels or by elevating NaMN or NaAD. (a, b) Metabolites were analyzed from control or NMN deamidase (NMN DD)-expressing neurons. Nicotinic acid mononucleotide (NaMN) and nicotinic acid adenine dinucleotide (NaAD) levels were normalized to those present in NMN DD-expressing neurons. NMN and $NAD^+$ levels were normalized to that of control neurons. Nicotinamide riboside (NR, 5 mM) was added at 24 hr prior to metabolite measurements. Data show the first and third quartile (box height) and median (line in the box) ± 1.5 time interquartile (details in the method), one-way ANOVA (NaMN, $F(4, 40) = 71.8$, $p<2\times10^{-16}$; NMN, $F(4, 40) = 72.69$, $p=1.64\times10^{-9}$; NaAD, $F(4,40) = 101.3$, $p < 2\times10^{-16}$; $NAD^+$, $F(4,40) = 121.4$, $p < 2 \times 10^{-16}$). *$p <1 \times 10^{-10}$ denotes significant difference from control (for NMN and $NAD^+$) or NMN DD-expressing (for NaMN or NaAD) metabolite levels with Holm- Bonferroni multiple comparison (*Figure 3—source data 1*, n = 9). (c) Axon degeneration index (DI) of NMN DD and/or NMN synthetase (NMN SN)-expressing neurons from 0 to 72 hr post axotomy; data show the first and third quartile (box height) and median (line in the box) ± 1.5 time interquartile (details in the method). One-way ANOVA ($F(24, 200) = 75.06$, $p<2 \times 10^{-16}$. *$p<0.002$, **$p<2 \times 10^{-16}$denote significant difference from control DI before axotomy (*Figure 3—source data 1*, n = 9).

The following source data is available for figure 3:

**Source data 1.** Steady state cellular NMN, NaMN, $NAD^+$, NaAD levels in control and NMN deamidase expressing neurons with or without NMN synthetase or NMN synthetase + NR (*Figure 3a,b*).

steady-state NAD$^+$ (summation of heavy and light NAD$^+$) was constant due to ongoing and balanced NAD$^+$ synthesis (6.9 ± 1.4% per hour) and NAD$^+$ consumption (8.2 ± 3.1% per hour) (*Figure 4a*). Next we compared basal NAD$^+$ flux in uninjured axons under various axoprotective conditions. NAD$^+$ biosynthesis enzymes were expressed and axonal NAD$^+$ consumption rates were measured at 4 hr post D4-Nam addition. Absolute NAD$^+$ consumption rates were normalized to ATP levels, which are constant in uninjured axons. We found that axon NAD$^+$ consumption in uninjured axons was similar to wild type levels in SARM1 KO neurons as well as neurons overexpressing cytNMNAT1. NAD$^+$ consumption was lower in axons from NMN deamidase-expressing neurons and higher in axons from NAMPT-expressing neurons (*Figure 4—figure supplement 3* and and *Figure 4—source data 2*). Since NAD$^+$ levels are at equilibrium in these uninjured axons, these changes in NAD$^+$ consumption are balanced by equivalent changes in NAD$^+$ production. Hence, basal levels of NAD$^+$ flux are well correlated with steady-state NAD$^+$ levels (lower in NMN deamidase-expressing neurons and higher in NAMPT-expressing neurons) but not with the axonal protection phenotypes (*Figure 1c* and *Figure 1—source data 1*, *Figure 2b,c* and *Figure 2—source data 1*, *Figure 4—figure supplement 3*). These results are expected since NMN deamidase inhibits NAD$^+$ synthesis by reducing the levels of NMN whereas NAMPT promotes NAD$^+$ synthesis by increasing the levels of NMN. NMN is the rate limiting metabolite in the NAD$^+$biosynthetic pathway (*Revollo et al., 2004*).

To analyze NAD$^+$ flux following injury, D4-Nam was added at the time of axotomy. In axotomized axons, steady-state NAD$^+$ levels declined as previously reported. We now show that this is due to a large increase in the NAD$^+$ consumption rate (from 8.5 ± 3.8% per hour prior to injury to 21.7 ± 1.6% per hour after injury: *Figure 4b* and *Figure 4—source data 1*, *Figure 4—figure supplement 4*). Surprisingly, in the first two hours after injury, there was also a modest increase in NAD$^+$ biosynthesis (control: 6.6 ± 3.7% per hour vs. post-axotomy: 14.0 ± 6.9% per hour; *Figure 4a, c* and *Figure 4—source data 1*). This is likely due to the activation of NAMPT, the rate-limiting enzyme in the NAD$^+$ biosynthetic pathway, which is negatively regulated by NAD$^+$. The decline in NAD$^+$ relieves this feedback inhibition and results in increased NAMPT activity (*Elliott and Rechsteiner, 1982*), which is also the likely explanation for the transient increase in axonal NMN after injury (*Figure 2d* and *Figure 2—source data 1*). Although the NAD$^+$ synthesis rate was increased during the first two hours after axotomy (*Figure 4c* and *Figure 4—source data 1*), total NAD$^+$ declined due to the larger increase in the NAD$^+$ consumption rate (*Figure 4a*). After two hours post-axotomy, heavy NAD$^+$ declined at the same rate as light NAD$^+$ (*Figure 4a*), indicating a complete loss of NAD$^+$ biosynthesis, likely due to loss of the labile NMNAT2 in the severed axon (*Gilley et al., 2010*). Hence, flux analysis demonstrates that injury-induced NAD$^+$ depletion is due to a major increase in NAD$^+$ consumption that begins shortly after the axon damage and before the loss of axonal NAD$^+$ synthesis (*Figure 4c* and *Figure 4—source data 1*). This surprising finding highlights the utility of flux analysis to provide mechanistic insights that were not appreciated in prior studies of steady-state NAD$^+$ levels.

We hypothesized that SARM1 may be required for this injury-dependent increase in NAD$^+$ consumption because we previously demonstrated that SARM1 is required for the steady-state loss of NAD$^+$ following injury in vivo and that SARM1 activation in primary cultured neurons leads to NAD$^+$ degradation (*Gerdts et al., 2015*). To test this hypothesis, we applied NAD$^+$ flux analysis to SARM1-deficient neurons. The NAD$^+$ consumption rate in SARM1-deficient axons was comparable to control axons prior to injury (*Figure 4a*, *Figure 4—figure supplement 3* and *Figure 4—source data 2*). However in contrast to the dramatic increase in NAD$^+$ consumption in wildtype neurons after injury, there was no significant change in NAD$^+$ consumption rate after injury in SARM1 mutant axons (uninjured: 6.3 ± 2.4% per hour vs. SARM1 mutant: 7.9 ± 3.7% per hour after axotomy). Hence, SARM1 is required for the injury-induced increase in NAD$^+$ consumption (*Figure 4a,b* and *Figure 4—source data 1*, *Figure 4—figure supplement 4*). This supports and extends our previous finding that SARM1 is responsible for initiating events that lead to the depletion of axonal NAD$^+$ following axotomy (*Gerdts et al., 2015*).

Since SARM1 is required for axon degeneration and axonal NAD$^+$ depletion, we next assessed whether other manipulations that affect axon survival do so by blocking injury-induced SARM1-dependent axonal NAD$^+$ depletion (*Figure 4b* and *Figure 4—source data 1*). The overexpression of NAMPT, which leads to modest axon protection, did not block injury-induced SARM1-dependent NAD$^+$ consumption. In contrast, expression of either cytNMNAT1 or NMN deamidase completely eliminated SARM1-dependent NAD$^+$ consumption after axotomy (*Figure 4b* and *Figure 4—source*

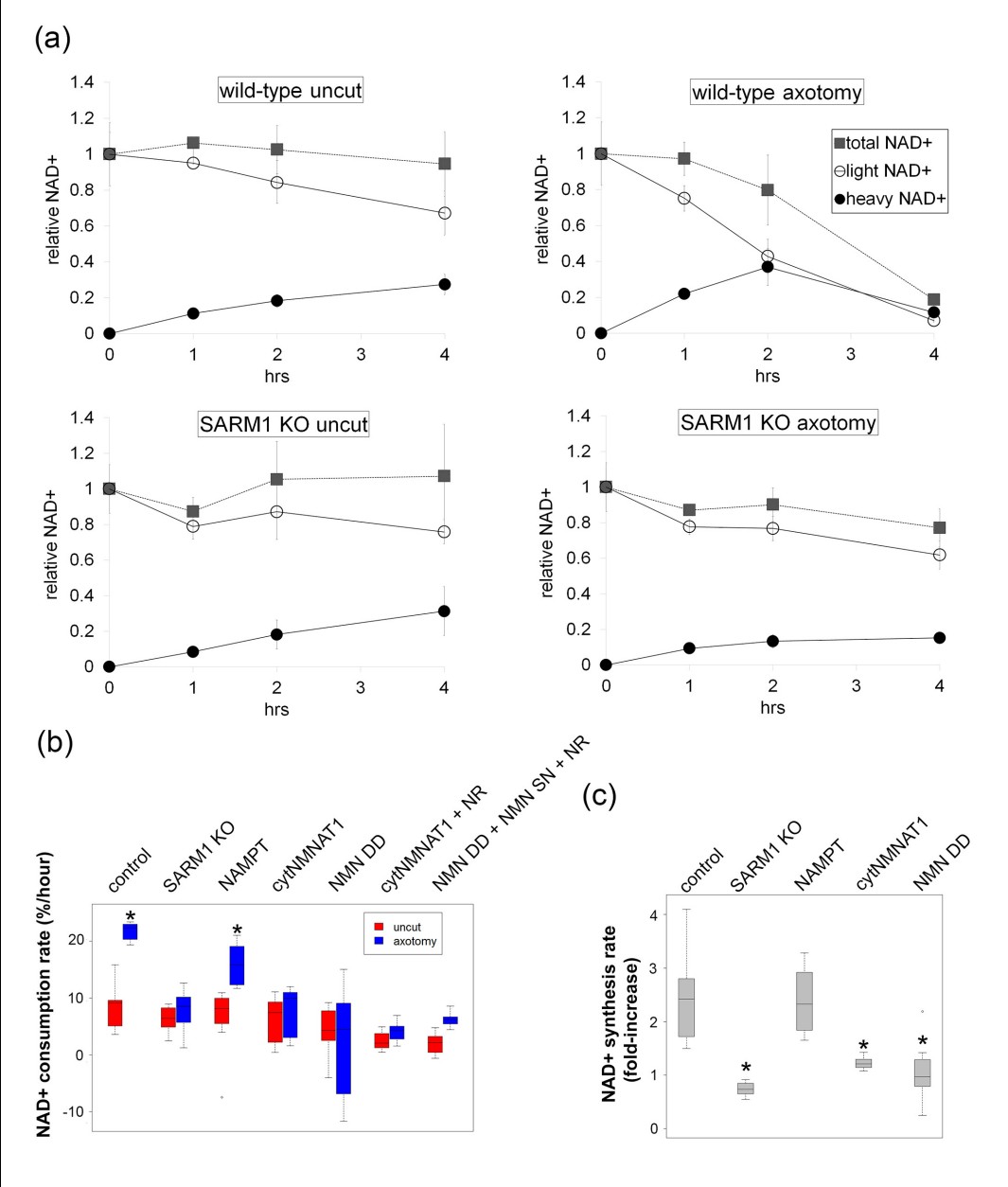

**Figure 4.** SARM1-dependent injury-induced NAD$^+$ consumption in the axon is the critical determinant of axon degeneration. (a) Axonal NAD$^+$ consumption and synthesis rates were measured by using stable isotope Nam (D4-Nam; 2,4,5,6-deutrium Nam). Culture medium was supplemented with 300 μM of D4-Nam at time 0 and then deuterium-labeled (heavy) NAD$^+$ (to follow synthesis) and non-labeled (light) NAD$^+$ (to follow consumption) were measured from axonal metabolites using LC-MS/MS. Representative graphs of NAD$^+$ synthesis and consumption before and after axotomy in wt and SARM1 KO are shown (mean ± s.d., n=3 for each data point). (b) Quantification of axonal NAD$^+$ consumption rate under various manipulations of NAD$^+$ biosynthesis pathways before and 4 hr after axotomy (NR concentration was 1 mM for cytNMNAT1 and 5 mM for NMN DD NMN SN expressing cells); data show the first and third quartile (box height) and median (line in the box) ± 1.5 time interquartile (details in the method), one-way ANOVA $F_{(13,118)}$ = 17.19 p<2 × 10$^{-16}$. *p<0.005 denotes significant difference from the NAD$^+$ consumption rate of control before axotomy with Holm-Bonferroni multiple comparison (*Figure 4—source data 1*, n=9 except NAMPT where n = 12). (c) Fold increase of axonal NAD$^+$ synthesis at 2 hr after axotomy compared with the uncut control under various manipulations of the NAD$^+$ biosynthesis pathway. The NAD$^+$ synthesis rate was normalized to that of corresponding uncut axons; data show the first and third quartile (box height) and median (line in the box) ± 1.5 time interquartile (details in the method), one-way ANOVA $F_{(4, 43)}$ = 21.41, p=9 × 10$^{-10}$. *p<4 × 10$^{-5}$ denotes a significant difference form fold increase of NAD$^+$ synthesis in

*Figure 4 continued on next page*

*Figure 4 continued*

control with Holm-Bonferroni multiple comparison (*Figure 4—source data 1*, n=9 except for control where n = 12).

The following source data and figure supplements are available for figure 4:

**Source data 1.** Relative axonal NAD⁺ consumption (*Figure 4b*) and synthesis (*Figure 4c*) rates in axons expressing various NAD⁺ biosynthesis enzymes with or without NR at 0, 4 hr (consumption), or 2 hr (synthesis) after axotomy.

**Source data 2.** Axonal degeneration index at 0, 9 and 24 hr post axotomy in the presence or absence of 1 mM Nicotinamide (*Figure 4—figure supplement 1*).

**Source data 3.** Absolute Axonal NAD⁺ consumption rates in axons expressing various NAD⁺ biosynthesis enzymes (*Figure 4—figure supplement 3*).

**Figure supplement 1.** Axonal degeneration profiles after axotomy were not altered by the addition of 1 mM Nam to the culture medium at the time of axotomy (*Figure 4—source data 2*, n=4).

**Figure supplement 2.** Representative NAD⁺ consumption and synthesis in DRG neurons.

**Figure supplement 3.** Absolute axonal NAD⁺ consumption rates in neurons expressing various NAD⁺ biosynthesis enzymes or in control neurons.

**Figure supplement 4.** Representative axonal NAD⁺ consumption under various manipulations of NAD⁺ biosynthesis pathways are shown (mean ± s.d., n=3 for each data point).

*data 1*). This is a striking result that likely explains the profound axoprotective effects of overexpressing either NMNAT or NMN deamidase — both block the ability of SARM1 to trigger NAD⁺ degradation following axotomy. Loss of SARM1 or gain of NMNAT or NMN deamidase all lead to profound and long-lasting axonal protection following injury, and all three block NAD⁺ degradation following injury. In contrast, treatments that provide modest axon protection (e.g. NAMPT) likely do so by transiently countering injury-induced NAD⁺ degradation through their ability to elevate intracellular NAD⁺ and thus maintain axonal NAD⁺ levels above a critical threshold for a longer period after injury.

We have shown that a reduction in NMN is not required to block axon degeneration (*Figure 1–3* and *Figure 3—source data 1*), however the role of NMN in SARM1 activation has not been investigated. Using injury-activated NAD⁺ -consumption as an assay of SARM1 pathway activity, we tested whether maintaining low NMN levels is necessary for the blockade of SARM1 pathway activation by cytNMNAT1 or NMN deamidase. Intracellular NMN was raised by the addition of NR to cytNMNAT1 expressing neurons (5.3 ± 2.9 fold) or NMN deamidase and NMN synthetase expressing neurons (3.7 ± 3.1-fold) and SARM1-dependent NAD⁺ consumption was measured. In both cases, SARM1-triggered

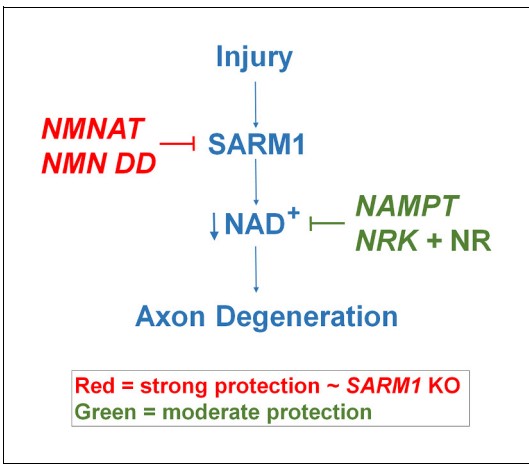

**Figure 5.** Schematic representation of molecular mechanism of axon degeneration. Axonal injury activates SARM1 that then results in an increase in NAD⁺ consumption that is soon followed by axon fragmentation (blue). NMNAT enzymes and NMN deamidase inhibit SARM1 induction of NAD⁺ consumption and provide strong axonal protection that is indistinguishable from SARM1-deficient axons (red). NAMPT or NRK + NR raise intracellular NAD⁺ in pre-injured axons thus delaying the point where NAD⁺ levels reach a critical threshold and axonal degeneration ensues (green). The axonal protection mediated by NAMPT or NRK + NR is not as long-lasting as that mediated by NMNAT enzymes or NMN deamidase.

axonal NAD$^+$ degradation after injury is fully inhibited by cytNMNAT1 or NMN deamidase despite high levels of NMN (*Figure 4b* and *Figure 4—source data 1*). Hence, a reduction in NMN is not necessary for blockade of SARM1 pathway activation by NMNAT1 or NMN deamidase.

These findings support a model in which axotomy activates SARM1, which then triggers events that lead to a catastrophic loss of NAD$^+$ followed by axon degeneration. Here we identify two distinct mechanisms that block axon degeneration (*Figure 5*). The most potent protection comes from overexpression of cytNMNAT1 or NMN deamidase, both of which block SARM1-dependent NAD$^+$ depletion in the injured axon. By inference, NMNAT2 also works via this mechanism, thus when its levels decline in the injured axon, SARM1 becomes activated. This biochemical demonstration that NMNAT enzymes inhibit SARM1 activation is consistent with a recent study showing that the embryonic lethality of NMNAT2 knockout mice is rescued by the loss of SARM1. This provides strong genetic evidence that NMNAT2 functions upstream of SARM1 (*Gilley et al, 2015*, *Figure 5*). Weaker axonal protection is provided by overexpression of NAMPT or by treating neurons expressing NRK1 with NR. These latter manipulations do not block SARM1-dependent NAD$^+$ consumption, but do lead to a large increase in steady-state NAD$^+$ levels prior to axotomy that likely maintains NAD$^+$ above a critical threshold for an extended period of time after SARM1 activation (*Figure 5*). As in other studies (*Sasaki et al., 2009b*, *Di Stefano et al., 2015*), we also observed that FK866, a NAMPT inhibitor, delays axon degeneration. We do not have an adequate explanation for this phenomenon; however, it should be noted that FK866 provides much weaker protection than that afforded by NAMPT (*Figure 1*, and *Figure 1—source data 1*).

In this work we demonstrate that blockade of SARM1 activation and axonal protection is not due to an increase in NAD$^+$ or a decrease in NMN. How then do NMNAT enzymes (and NMN deamidase) block SARM1-stimulated axon degeneration and NAD$^+$ loss? One possibility is that these enzymes physically interact with SARM1 to inhibit its ability to activate axon degeneration pathways including those involved in NAD$^+$ loss. However, NMNAT enzymes from many species, including the archean *Methanocaldococcus jannaschii*, provide dramatic axonal protection despite their minimal amino acid sequence similarity, thus making direct interaction with SARM1 an unlikely mechanism (*Sasaki et al., 2009b*). Second, the necessity for increased NAD$^+$ levels in a specific axonal compartment is an attractive explanation, however there is no evidence for this mechanism. Protein transduction experiments with cytNMNAT1 show rapid and strong axon protection, and it seems unlikely that a mutant nuclear protein would end up in a specific axonal compartment necessary to promote axon protection (*Sasaki et al., 2009a*). Finally, mutations in both NMNAT enzymes and NMN deamidase that destroy their enzymatic activity also block their axon protective capabilities (*Araki et al., 2004*, *Sasaki et al., 2009b*, *Di Stefano et al., 2015*). This strongly suggests that metabolites whose levels are altered by NMNAT and/or NMN deamidase activity regulate SARM1 function. NMN was such a candidate metabolite because it is a substrate for both NMNAT enzymes and NMN deamidase; however, our studies show that NMN levels do not regulate NAD$^+$ loss and axon degeneration, both SARM1-induced activities. Further investigations to identify other cellular metabolites modulated by NMNAT or NMN deamidase may help elucidate the mechanism of SARM1 regulation and axonal degeneration.

These studies support the hypothesis that SARM1-dependent NAD$^+$ consumption is the central biochemical event in the axonal degeneration program. Further, they explain the phenotype of the *wlds* mutant mouse as one caused by the continued inhibition of SARM1 by the longer-lived Wlds protein that compensates for the loss of NMNAT2. Finally, these findings suggest that inhibitors of SARM1 or agents that boost NAD$^+$synthesis are likely to be valuable approaches for blocking axonal degeneration in the injured or diseased nervous system.

## Materials and methods

### Mouse

CD1 mice (gestation day 11.5; Charles River Laboratories) and SARM1 knockout mice (C57/BL6; *Szretter et al., 2009*) are housed (12 hr dark/light cycle and less than 5 mice per cage) and used under the direction of institutional animal study guidelines at Washington University in St. Louis.

## Chemicals

Nicotinamide riboside (NR) was a gift from ChromaDex, Inc. Deuterium labeled nicotinamide (D4-Nam: 2,3,4,5 deuterium Nam) was obtained from C/D/N Isotopes Inc. NR and D4-Nam were dissolved in water at concentration of 100 mM and stored at –20℃. FK866 was obtained from National Institute of Mental Health Chemical Synthesis and Drug Supply Program (MH number F-901), solubilized in DMSO (final concentration 100 mM) and stored at −20℃. Nicotinamide mononucleotide (NMN), nicotinicacid mononucleotide (NaMN), nicotinamide adenine dinucleotide ($NAD^+$), and nicotinicacid adenine dinucleotide (NaAD) were all obtained from Sigma and stored at 20℃ as 100 mM solutions in water.

## cDNA

Lentivirus transfer vector constructs harboring cDNAs including mCherry, cytNMNAT1 (*mouse*), NAMPT (*mouse*), NRK1 (*mouse*) were previously described (*Sasaki et al., 2006*). For NMN deamidase (*E. Coli*) and NMN synthetase (*F. tularensis*), mammalian codon optimized double strand DNA of each coding region with 6xHis tag was synthesized (gBlocks gene fragments; Invitrogen) and cloned into lentivirus transfer vector FCIV (*Araki et al., 2004*) derived from FUGW (a gift from David Baltimore; Addgene plasmid #14883; *Lois et al., 2002*) using InFusion (Clontech). The DNA sequences for NMN deamidase and NMN synthetase are presented below.

### NMN deamidase

5′ ATGACCGACTCCGAGCTGATGCAGTTGTCTGAGCAGGTGGGCCAGGCATTGAAGGCAA-GAGGGGCTACAGTCACAACCGCTGAGTCCTGTACTGGCGGATGGGTGGCAAAGGTCATAACTGA TATTGCCGGATCAAGCGCCTGGTTTGAGAGAGGCTTCGTGACATATTCAAATGAAGCAAAAGCC-CAAATGATTGGCGTTCGAGAGGAAACCCTCGCTCAACACGGAGCTGTAAGTGAGCCTGTTGTAG TGGAAATGGCAATCGGTGCCCTTAAAGCTGCAAGGGCTGATTACGCAGTATCCATTTCAGGCA TAGCAGGTCCCGATGGCGGATCAGAGGAGAAACCTGTCGGGACTGTATGGTTCGCCTTTGCAA-CAGCACGGGGAGAAGGGATAACCCGAAGAGAATGTTTTTCTGGTGATCGGGATGCTG TGCGACGGCAGGCCACCGCCTATGCTCTTCAAACTCTCTGGCAGCAGTTCCTCCAAAATACA<u>CA TCACCATCATCATCAT</u>TTAG-3′
 6xHis-tag

### NMN synthetase

5′-ATGAAGATTGTGAAGGACTTTAGCCCCAAAGAGTATAGTCAGAAACTTGTTAATTGGTTGAG TGACAGCTGTATGAATTACCCTGCTGAGGGATTCGTGATAGGTCTCTCAGGAGGCATCGATAG TGCCGTCGCTGCCTCTCGCCGTCAAGACTGGACTGCCAACCACTGCCCTTATTCTGCCATC TGACAACAATCAGCACCAGGACATGCAGGACGCCCTCGAGCTGATCGAAATGTTGAACA TAGAGCATTACACAATCTCAATTCAGCCTGCCTACGAAGCCTTTCTTGCTTCTACTCAGAGCTTCA-CAAACCTGCAGAATAACCGGCAGCTGGTGATTAAAGGGAATGCTCAGGCACGTCTCCGCATGA TGTATCTGTACGCATATGCACAGCAGTATAACCGGATCGTTATCGGGACAGATAACGCCTGCGAG TGGTATATGGGCTACTTTACCAAGTTCGGGGATGGTGCCGCTGACATCCTCCCTCTGGTGAATC TGAAGAAATCCCAGGTTTTTGAGTTtGGAAAATACCTGGACGTGCCAAAAAACATTCTGGA-CAAGGCACCCTCCGCTGGCCTTTGGCAGGGCCAGACAGATGAAGATGAAATGGGCGTGACC TACCAGGAAATAGACGATTTCCTGGATGGGAAGCAGGTCTCCGCAAAGGCACTGGAGCGAA TCAACTTCTGGCATAACAGAAGCCACCACAAGAGGAAATTGGCTCTCACCCCCAATTTT<u>CA TCACCATCATCATCAT</u>TGA -3′
 6xHis-tag

## DRG drop culture

Mouse DRG culture was performed as described before (*Sasaki et al., 2009b*). Mouse dorsal root ganglion (DRG) was dissected from embryonic days 13.5 CD1 mouse embryo (50 ganglion per embryo) and incubated with 0.05% Trypsin solution containing 0.02% EDTA (Gibco) at 37℃ for 15 min. Then cell suspensions are triturated by gentle pipetting and washed 3 times with the DRG growth medium (Neurobasal medium (Gibco) containing 2% B27 (Invitrogen), 100 ng/ml 2.5S NGF (Harlan Biproduts), 1 µM uridine (Sigma), 1 µM 5-fluoro-2′-deoxyuridine (Sigma), penicillin, and streptomycin). Cells were suspended in DRG growth medium at a ratio of 100 µl medium/50 DRGs.

The cell density of these suspensions was ~7x10⁶ cells/ml. Cell suspensions (1.5 µl/96 well, 10 µl/24 well) were placed in the center of the well using either 96- or 24-well tissue culture plates (Corning) coated with poly-D-Lysine (0.1 mg/ml; Sigma) and laminin (3 µg/ml; Invitrogen). Cells were allowed to adhere in humidified tissue culture incubator (5% $CO_2$) for 15 min and then DRG growth medium was gently added (100 µl/96 well, 500 µl/24 well). Lentiviruses were added (1-10×10³pfu) at 1–2 days in vitro (DIV) and metabolites were extracted or axon degeneration assays were performed at 6–7 DIV. When using 24 well DRG cultures, 50% of the medium was exchanged for a fresh medium at DIV4. NR (100 µM) was added 24 hr before axotomy or metabolite collection.

## Lentivirus

Lentiviruses were produced in HEK293T cells (RRID: CVCL_0063) as previously described (*Araki et al., 2004*). Cells were seeded at a density of $1 \times 10^6$ cells per 35 mm well the day before transfection. FCIV lentivirus constructs harboring cDNAs encoding mCherry, cytNMNAT1 (*mouse*), NAMPT (*mouse*), NRK1 (*mouse*), NMN deamidase (*E. Coli*), or NMN synthetase (*F. tularensis*) (each 400 ng) was co-transfected with VSV-G (400 ng) and pSPAX2 (1.2 µg) using X-tremeGene (Roche). The virus containing medium was collected at 3–5 days after transfection. The lentivirus particles (1–10 x 10⁶ infectious particles/ml) were collected from the cleared culture supernatant with Lenti-X concentrator (Clontech), suspended in 1/10 original volume of PBS, and stored at −80°C.

## Metabolite collection

At DIV6, the 24-well DRG culture plate was placed on ice and the medium was replaced with ice-cold 0.9% NaCl solution. DRG metabolites were extracted using ice-cold 1:1 mixture of MeOH and water (150 µl per well) on ice for 10 min. For axonal metabolite collection, neuronal cell bodies and axons were separated using a microsurgical blade under the microscope. The culture medium was replaced with saline and the DRG cell bodies were removed prior to metabolite extraction using a pipet. The metabolite containing solutions were transferred into test tubes and extracted twice with chloroform (100 µl per sample). The aqueous phase (120 µl) was lyophilized and stored at −20°C until analysis.

## Metabolite measurement using LC-MS/MS

Lyophilized samples were reconstituted with 50 µl of 5 mM ammonium formate and centrifuged at 12,000 x g for 10 min. Cleared supernatant were transferred to sample vials. Serial dilutions of standards for each metabolite in 5 mM ammonium formate were used for calibration. The highest standard concentration used were 25 µM for NAD⁺, nicotinicacid adenine dinucleotide (NaAD), nicotinamide mononucleotide (NMN), ATP, and ADP, 5 µM for nicotinicacid mononucleotide (NaMN), and 1 µM for AMP. Liquid chromatography was performed by HPLC (1290; Agilent) with Atlantis T3 (LC 2.1 x 150 mm, 3 µm; Waters) for steady-state metabolite assays (*Hikosaka et al., 2014*) and Synergi Fusion-RP (4.6 x 150 mm, 4 µm; Phenomenex) for NAD⁺ flux assays. For steady-state metabolite analysis, 20 µl of samples were injected at a flow rate of 0.15 ml/min with 5 mM ammonium formate for mobile phase A and 100% methanol for mobile phase B. Metabolites were eluted with gradients of 0–10 min, 0–70% B; 10–15 min, 70% B; 16–20 min, 0% B (*Hikosaka et al., 2014*). For NAD⁺ flux assay, 10 µl of each sample were injected at a flow rate of 0.55 ml/min with 5 m ammonium formate for mobile phase A and 100% methanol for mobile phase B. Metabolites were eluted with gradients of 0–7 min, 0–70% B; 7–8 min, 70% B; 9–12 min, 0% B. The metabolites were detected with a Triple Quad mass spectrometer (6460 MassHunter; Agilent) under positive ESI multiple reaction monitoring (MRM) using parameters for each compound as shown in the *Table 1*. Metabolites were quantified by MassHunter quantitative analysis tool (Agilent) with standard curves and relative metabolite concentrations compared with control neurons (for NMN, NAD⁺) or with NMN deamidase-expressing neurons (for NaMN, NaAD) were calculated.

## NAD⁺ flux assay

For NAD⁺ consumption and synthesis measurements, DRG neurons were incubated with D4-Nam (300 µM: 2,3,4,5 deuterium Nam; C/D/N Isotopes Inc.) for 0–4 hr and axonal metabolites were collected as described above. NAD⁺ synthesis and consumption rates were measured at 2 and 4 hr post D4-Nam addition, respectively. For NAD⁺ flux measurements after axonal injury, D4-Nam was

**Table 1.** Mass spectroscopy parameters for metabolites measured.

| Metabolite | MS1/MS2 | Fragmentation (V) | Collision energy (V) | Cell AC (V) |
| --- | --- | --- | --- | --- |
| NAD$^+$ | 664>428 | 160 | 22 | 7 |
| D4-NAD$^+$ | 668>428 | 160 | 22 | 7 |
| D3-NAD$^+$ | 667>428 | 160 | 22 | 7 |
| NaAD | 669>428 | 150 | 20 | 7 |
| NMN | 335>123 | 90 | 11 | 1 |
| NaMN | 336>124 | 80 | 10 | 7 |
| ATP | 508>136 | 90 | 30 | 2 |
| ADP | 428>136 | 50 | 30 | 3 |
| AMP | 348>136 | 30 | 30 | 7 |

added at the same time as axotomy. Labeled (heavy) or non-labeled (light) NAD$^+$ was quantified as described below with LC-MS/MS. For heavy-labeled NAD$^+$, D3-NAD$^+$ as well as D4-NAD$^+$ was observed as previously reported (*Hara et al., 2014*). This is due to the replacement of deuterium at C4 position with non-labeled proton during NAD$^+$d this combined value-NADH cycling. We added the values of D3-NAD$^+$ and D4-NAD$^+$and used this combined value as the amount of newly synthesized NAD$^+$. The net rate of NAD$^+$ synthesis and consumption were calculated by% increase or decrease of metabolite/hour at 2 hr (synthesis) or 4 hr (consumption) after D4-Nam application. Metabolite concentrations from 9 to 12 independent wells were compared using one-way ANOVA and Holm-Bonferroni multiple comparison using R (RRID:SCR_002394).

## Axon degeneration assay

Axons from DRG drop cultures in 96 well were transected using a micro surgical blade under microscope at DIV6. Bright field images of distal axons (6 fields per well) were taken at 0–72 hr after axotomy using a high content imager (Operetta; Perkinelmer) with 20x objective (*Gerdts et al., 2011*). Axon degeneration was quantified using degeneration index (DI) calculated using ImageJ (NIH) as previously described (*Gerdts et al., 2011*; *Sasaki et al., 2009b*). The average DI from 6 fields per well was obtained and averaged for each independent well. The DI was calculated from axon images from the same fields before (0 hr) and after (9–72 hr) axotomy. Experiments were repeated 3 times with 3 independent wells (n=9). For statistical analysis DI was compared using one-way ANOVA and Holm-Bonferroni multiple comparison using R (RRID:SCR_002394).

## Statistics

Sample number (n) was defined as the number of cell culture wells that were independently manipulated and measured. No statistical evaluations were performed to predetermine sample sizes, but our sample sizes are similar to those generally used in the field. Data distribution was assumed to be normal, but this was not formally tested. Data comparisons were performed using one-way ANOVA using R (RRID:SCR_002394) and F and P values were reported for each comparison in corresponding figure legends. For multiple comparisons, the Holm-Bonferroni multiple comparison method was used. Statistical significance was noted as *, **, or # in each graph with indicated values in the corresponding legends. For two group comparison, unpaired Student t-test was performed using R.

Bar graph and line plot data are mean +/- S.D.. Box-plot was provided using R. The box height represents the first (bottom) and third (top) quartiles, and the median is indicated as a horizontal line in the box, and the top error bar represents the smaller of 1.5 times interquartile range or maximum data point and bottom error bar represents 1.5 times interquartile range or minimum data point. Any data beyond 1.5 times interquartile (25% to 75% quartile) are shown as open circle dots.

## Acknowledgement

This work was supported by the National Institutes of Health (Grant RO1AG013730 (JM), RO1NS065053 and RO1NS087632 (JM and AD)), National Institute of General Medical Science (8 P41 GM103422) from the National Institutes of Health, and JSPS (KAKENHI 24609019 (TN)). Nicotinamide riboside (NR) was a gift from ChromaDex (Irvine, California). We thank members of the Milbrandt and DiAntonio laboratories and Jun Yoshino for fruitful discussions. We thank T Farhner, K Kruse, N Panchenko, K Simburger, and A Strickland for experimental assistance.

## Additional information

### Competing interests

YS, JM: May derive benefit from a licensing agreement with ChromaDex, which did not provide any support for this work. The other authors declare that no competing interests exist.

### Funding

| Funder | Grant reference number | Author |
| --- | --- | --- |
| Japan Society for the Promotion of Science | KAKENHI 24609019 | Takashi Nakagawa |
| National Institutes of Health | RO1NS065053 | Aaron DiAntonio Jeffrey Milbrandt |
| National Institutes of Health | RO1NS087632 | Aaron DiAntonio Jeffrey Milbrandt |
| National Institutes of Health | RO1AG013730 | Jeffrey Milbrandt |

The funders had no role in study design, data collection and interpretation, or the decision to submit the work for publication.

### Author contributions

YS, Designed and performed experiments and analyzed data and prepared the manuscript; TN, Designed and performed experiments and analyzed data; XM, Designed and performed experiments; AD, JM, Designed and supervise experiments and prepared the manuscript, Analysis and interpretation of data

### Author ORCIDs

Yo Sasaki, http://orcid.org/0000-0003-0024-0031
Aaron DiAntonio, http://orcid.org/0000-0002-7262-0968

### Ethics

Animal experimentation: The experiments using animals in this study were performed in accordance with the recommendation in the Guide for the Care and Use of Laboratory Animals of the National Institute of Health. Animals were handled under the direction of institutional animal study guidelines at Washington University in St. Louis (protocol number: 20140044).

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
