## [Decision Letter]

Thank you for submitting your article "NMNAT1 inhibits axon degeneration via blockade of SARM1-mediated NAD+ depletion" for consideration by *eLife*. Your article has been reviewed by three peer reviewers, and the evaluation has been overseen by a Reviewing Editor and Gary Westbrook as the Senior Editor. The following individual involved in review of your submission has agreed to reveal his identity: Avraham Yaron (Reviewer #2). The reviewers have discussed the reviews with one another and the Reviewing Editor has drafted this decision to help you prepare a revised submission.

Summary:

The three referees were all knowledgeable about mechanisms of axonal regeneration, and made a number of positive comments about your manuscript with regard to the action of the NAD+ biosynthetic enzyme, NMNAT1 in axons. Each noted the ability to measure the flux of NAD+ metabolites as a strength of the manuscript. However, there were several concerns discussed by the reviewers and editor that require attention. Specifically,

1) The involvement of NAD+ turnover by SARM1 is regarded as an interesting and important finding, but more consideration concerning the role of NAD+ is required. In particular, further explanation of the mechanisms of axonal protection and how NMNAT1 blocks NAD+ utilization by SARM1 is needed.

2) The reviewers indicated that more clarity is needed to define the nomenclature and many acronyms (NMN, NaMN, NaAD etc.) to the non cognoscenti. Additional models or schematic diagrams would be particularly useful, as well as clarification of Figure 1.

3) There were also experimental issues that should be addressed, including (a) presentation of data regarding the quantification of NAD+ and the need for non-normalized data; (b) the changes in the lesion experiment of Figure 3; (c) as well as information on protein expression data in axons.

We think that careful attention to these specific issues as well as the other major concerns outlined below in the reviews would strengthen the study.

*Reviewer #1:*

Sasaki and colleagues present a detailed and rigorous analysis of NAD+ metabolism during axon injury and degeneration. The studies are motivated by the prior discovery that manipulating NAD+ metabolism slows Wallerian degeneration of cut axons. In this paper, the authors demonstrate that altered steady-state levels of NAD+ or its precursors does not per se induce degeneration and they establish a method to measure changes caused axotomy in the rates of NAD+ synthesis and consumption. Using this assay, the authors find that axotomy triggers a surge of NAD+ consumption that requires the SARM TIR-containing adapter protein and is blocked by expression of the NAD+ biosynthetic enzyme NMNAT1.

Overall, the manuscript clearly presents a set of compelling and satisfying experiments that advance our understanding of a mechanism central to Wallerian degeneration. I enthusiastic about the work and only have a few questions/comments.

1) The authors conclude that SARM-regulated NAD+ consumption is strongly affected by NAD+ biosynthetic enzymes but this regulation is not simply a consequence of elevated NAD+ levels prior to axotomy. The data support this conclusion. However, the discussion of this point is a bit cursory and does not go far beyond a restatement of the facts. Given that this is an unexpected result, it should be discussed a bit more. How might this work? Are there distinct pools of NAD+ in axons that can only be accessed by a subset of the manipulations employed by the authors? Are some enzymes able to interact with the SARM pathway and others not? Inquiring minds would like to know what the authors think.

2) The authors present rates of NAD+ synthesis and consumption normalized to pre-axotomy levels. The data should be presented as absolute rates of synthesis and consumption to allow the reader to also compare baseline NAD+ metabolism between conditions.

3) I would be very curious to know how SARM mutation affects levels of NAD+ and its metabolic precursors. If feasible, the authors should include these data in Figure 1.

*Reviewer #2:*

The work from the DiAntonio and Milbrandt labs examines the dynamics of NAD+ metabolites in axotomized axons in vitro. Through multiple manipulations and new NAD+ flux analysis they come to two main conclusions: 1. Increase in NMN upon injury is not the trigger and is not essential for axonal degeneration. This is in contrast to a recent published study (Di Stefano. et al., 2015). 2. Axonal protection by NMNAT1 is generated by inhibition of the SARM dependent NAD+ degradation/consumption and not through counter synthesis of NAD+. The second conclusion is very interesting and in many aspects contradicts the assumption that NMNAT1 or Wlds protect against axonal degeneration by supplement of NAD+ purely through synthesis. It also suggests that targeting of the SARM dependent NAD+ degradation pathway may be more effective than stimulation of NAMPT activity.

I find the paper somehow not easy to read, and some of the results are puzzling. In general the authors should expand the text to better explain their findings, otherwise non-experts will get lost.

1) There is no protein expression data of the overexpressed constructs by axonal IF or WB, this is even more important when two constructs are presumably co-expressed.

2) There is axonal protection both by overexpression of NAMPT and its inhibitor, FK866, how the authors explain that?

3) Figure 3 – this is the key figure in the paper and important information is missing. The authors do not discuss the data of panel C in the text. I understand that control is un-cut why the authors decided to omit the control cut data from panel C? In B and C for how long the axons are cut?

*Reviewer #3:*

The authors present evidence that NMNAT1 blocks SARM1 induced depletion of NAD in isolated axons as opposed to maintaining NAD at overall higher levels (i.e., delaying decline in NAD following axotomy). They have developed NAD flux analyses here by loading cultures with D4-NAM. On the one hand, this seems like quite a technical advance for a metabolomics approach applied to isolated axons that has been lacking in the field. Similarly, showing that elevated NMN levels do not correlate with axon degeneration is an advance that overturns some reports in the literature. The manuscript is also overall well written and the central conclusion that NMNAT1 blocks SARM mediated NAD depletion is quite intriguing, so the findings appear to be relatively novel and could advance the field. However, the authors provide no clear insight into the mechanism of how NMNAT1 blocks SARM function other than its known role in generating NAD. From a mechanistic point of view it is puzzling that this occurs without affecting steady state NAD levels. It is not clear to me from the data how NMNAT1 or NMN deamidase could do this without altering NAD production. Do these proteins interact with SARM1 or directly inhibit its function? At the very least, I think that the authors could add some speculation to the Discussion paragraph(s), though data would make a more compelling case.

---

## [Author Response]

Summary:

*The three referees were all knowledgeable about mechanisms of axonal regeneration, and made a number of positive comments about your manuscript with regard to the action of the NAD+ biosynthetic enzyme, NMNAT1 in axons. Each noted the ability to measure the flux of NAD+ metabolites as a strength of the manuscript. However, there were several concerns discussed by the reviewers and editor that require attention. Specifically,*

*1) The involvement of NAD+ turnover by SARM1 is regarded as an interesting and important finding, but more consideration concerning the role of NAD+ is required. In particular, further explanation of the mechanisms of axonal protection and how NMNAT1 blocks NAD+ utilization by SARM1 is needed.*

The reviewers asked for additional discussion of how NMNAT1 might block the NAD^+^ depletion that is induced by SARM1. Reviewer one mentioned that “inquiring minds would like to know what the authors think” and reviewer three asked us to “add some speculation to the Discussion.”

We have now added a paragraph to the Discussion that critically assesses three hypotheses for how NMNAT1 and NMN deamidase might block SARM1 function.

*2) The reviewers indicated that more clarity is needed to define the nomenclature and many acronyms (NMN, NaMN, NaAD etc.) to the non cognoscenti. Additional models or schematic diagrams would be particularly useful, as well as clarification of Figure 1.*

We appreciate the suggestions. We have clarified Figure 1 by a) making it larger b) spelling out the names and explaining the function of the relevant enzymes and c) writing a much more clear figure legend. We have also added a schematic as Figure 5 that summarizes the main conclusions of the manuscript. Finally, we also split the original Figure 1 into two figures, which gives us more space for Figure 1 but also allows for greater clarity throughout the figures. As such, the numbering system of figures has changed from the original.

*3) There were also experimental issues that should be addressed, including (a) presentation of data regarding the quantification of NAD+ and the need for non-normalized data; (b) the changes in the lesion experiment of Figure 3; (c) as well as information on protein expression data in axons.*

We address these issues in detail below. Briefly, we a) present a new Figure 4—figure supplement 3 that presents absolute NAD^+^ consumption rates at baseline, b) show and discuss the control data in the current Figure 4 (original Figure 3) as described in our response to reviewer 2 below, and c) added a new Figure 1—figure supplement 1 showing the protein expression data in axons as requested.

*We think that careful attention to these specific issues as well as the other major concerns outlined below in the reviews would strengthen the study.*

*Reviewer #1:*

*Overall, the manuscript clearly presents a set of compelling and satisfying experiments that advance our understanding of a mechanism central to Wallerian degeneration. I enthusiastic about the work and only have a few questions/comments.*

*1) The authors conclude that SARM-regulated NAD+ consumption is strongly affected by NAD+ biosynthetic enzymes but this regulation is not simply a consequence of elevated NAD+ levels prior to axotomy. The data support this conclusion. However, the discussion of this point is a bit cursory and does not go far beyond a restatement of the facts. Given that this is an unexpected result, it should be discussed a bit more. How might this work? Are there distinct pools of NAD+ in axons that can only be accessed by a subset of the manipulations employed by the authors? Are some enzymes able to interact with the SARM pathway and others not? Inquiring minds would like to know what the authors think.*

We have now added a paragraph to the Discussion that addresses three potential hypotheses to explain the data, including those suggested by the reviewer.

*2) The authors present rates of NAD+ synthesis and consumption normalized to pre-axotomy levels. The data should be presented as absolute rates of synthesis and consumption to allow the reader to also compare baseline NAD+ metabolism between conditions.*

As requested by the reviewer, we have now added a Figure 4—figure supplement 3 that shows the absolute NAD^+^ consumption rate at baseline in the various genotypes studied in this manuscript. Because NAD^+^ is at steady state in uninjured axons, this consumption rate is equivalent to the synthesis rate and so we do not present both values. These results show that the baseline NAD^+^ flux is not correlated with axonal protection, but rather with the level of NAD^+^. These results are expected since the NMN deamidase inhibits NAD^+^ synthesis by reducing the levels of NMN and NAMPT promotes NAD^+^ synthesis by increasing the levels of NMN. NMN is the rate limiting metabolite in the NAD biosynthetic pathway. Wild type, SARM KO, and NMNAT1 have normal levels of NAD^+^ and normal NAD^+^ flux at baseline.

*3) I would be very curious to know how SARM mutation affects levels of NAD+ and its metabolic precursors. If feasible, the authors should include these data in Figure 1.*

We previously showed that steady-state NAD^+^ levels are unaltered in SARM1 KO vs. wildtype neurons in vitro and in vivo (Gerdts et al., 2015). We also show here that NAD^+^ flux rate is unaltered from wildtype in uninjured SARM1 KO neurons (Figure 4, Figure 4—figure supplement 3). These results suggest that NMN (the precursor of NAD^+^ synthesis) is also unchanged in SARM1 KO neurons. To confirm this, we have now measured NAD^+^ and its precursor metabolites using LC-MSMS in uninjured SARM1 KO and wildtype DRG neurons. The values are shown in Figure 2—figure supplement 1. There is no significant difference in the baseline levels of NMN, NaMN, NaAD, or NAD^+^ between SARM1 KO and wildtype neurons.

*Reviewer #2:*

*I find the paper somehow not easy to read, and some of the results are puzzling. In general the authors should expand the text to better explain their findings, otherwise non-experts will get lost.*

We appreciate the suggestion and have made numerous improvements throughout. As described above, we have added text, expanded the schematic in Figure 1 to describe the relevant enzymes more clearly, and added a new figure (Figure 5) that displays a schematic outlining our major findings. We believe these additions make the manuscript easier to follow.

*1) There is no protein expression data of the overexpressed constructs by axonal IF or WB, this is even more important when two constructs are presumably co-expressed.*

We performed Western blotting using axon lysates prepared from neurons expressing the indicated NAD^+^ biosynthesis enzymes (cytNMNAT1, NAMPT, NMN deamidase, NMN synthetase, NRK1, and a combination of NMN deamidase and synthetase) using an antibody to an epitope (His tag) present on all of the proteins tested. These data are shown in Figure 1—figure supplement 1.

*2) There is axonal protection both by overexpression of NAMPT and its inhibitor, FK866, how the authors explain that?*

This is a puzzling finding that has been reported previously (Sasaki et al., 2009b, Di Stefano et al., 2015) and was discussed in our recent review article (Gerdts et al.,2016). While in this paper we identify the mechanism by which NAMPT confers protection, we do not have an adequate explanation for the protection caused by FK866. We do note that the FK866 protection is much weaker than that provided by NAMPT expression (Figure 1). We now discuss this curious finding in the Discussion.

*3) Figure 3 – this is the key figure in the paper and important information is missing. The authors do not discuss the data of panel C in the text. I understand that control is un-cut why the authors decided to omit the control cut data from panel C? In B and C for how long the axons are cut?*

The original Figure 3 is now Figure 4. The reviewer was concerned that we did not discuss panel C in the original submission. We now extensively discuss Figure 4 in the text. The reviewer also asked about the uncut control in panel C. There data are included in panel C, because we present the data as the fold change in the NAD^+^ synthesis rate from before (uncut) to after (cut) injury. Finally, the reviewer asks how long the axons were cut in panels 4B and 4C. We calculated the NAD^+^ consumption rate (4B) at four hours after cut. We calculated the synthesis rate (4C) at two hours after cut. We used a shorter time for the synthesis rate because after two hours NAD^+^ synthesis essentially stops (as shown in Figure 4, and likely due to the previously described loss of Nmnat2 by about two hours after injury). We now include these times in the figure legend.

*Reviewer #3:*

*The authors present evidence that NMNAT1 blocks SARM1 induced depletion of NAD in isolated axons as opposed to maintaining NAD at overall higher levels (i.e., delaying decline in NAD following axotomy). They have developed NAD flux analyses here by loading cultures with D4-NAM. On the one hand, this seems like quite a technical advance for a metabolomics approach applied to isolated axons that has been lacking in the field. Similarly, showing that elevated NMN levels do not correlate with axon degeneration is an advance that overturns some reports in the literature. The manuscript is also overall well written and the central conclusion that NMNAT1 blocks SARM mediated NAD depletion is quite intriguing, so the findings appear to be relatively novel and could advance the field. However, the authors provide no clear insight into the mechanism of how NMNAT1 blocks SARM function other than its known role in generating NAD. From a mechanistic point of view it is puzzling that this occurs without affecting steady state NAD levels. It is not clear to me from the data how NMNAT1 or NMN deamidase could do this without altering NAD production. Do these proteins interact with SARM1 or directly inhibit its function? At the very least, I think that the authors could add some speculation to the Discussion paragraph(s), though data would make a more compelling case.*

As described in point one to reviewer one above, we have now added a paragraph to the Discussion that goes through three hypotheses to explain these new findings.